# Dietary Chitosan Attenuates High-Fat Diet-Induced Oxidative Stress, Apoptosis, and Inflammation in Nile Tilapia (*Oreochromis niloticus*) through Regulation of Nrf2/Kaep1 and Bcl-2/Bax Pathways

**DOI:** 10.3390/biology13070486

**Published:** 2024-06-30

**Authors:** Aya G. Rashwan, Doaa H. Assar, Abdallah S. Salah, Xiaolu Liu, Ibrahim I. Al-Hawary, Mohammed H. Abu-Alghayth, Shimaa M. R. Salem, Karim Khalil, Nemany A. N. Hanafy, Alaa Abdelatty, Luyang Sun, Zizy I. Elbialy

**Affiliations:** 1Department of Fish Processing and Biotechnology, Faculty of Aquatic and Fisheries Sciences, Kafrelsheikh University, Kafrelsheikh 33516, Egypt; aya.gamal@fsh.kfs.edu.eg (A.G.R.); hawaryib@fsh.kfs.edu.eg (I.I.A.-H.); 2Clinical Pathology Department, Faculty of Veterinary Medicine, Kafrelsheikh University, Kafrelsheikh 33516, Egypt; doaa.abdelhady1@vet.kfs.edu.eg; 3Department of Aquaculture, Faculty of Aquatic and Fisheries Sciences, Kafrelsheikh University, Kafrelsheikh 33516, Egypt; abdallah_salah_2014@fsh.kfs.edu.eg; 4Institute of Aquaculture, Faculty of Natural Sciences, University of Stirling, Stirling FK9 4LA, UK; 5Single-Cell Center, Shandong Key Laboratory of Energy Genetics and Shandong Energy Institute, Qingdao Institute of Bioenergy and Bioprocess Technology, CAS Key Laboratory of Biofuels, Chinese Academy of Sciences, Qingdao 266101, China; liuxl@qibebt.ac.cn; 6University of Chinese Academy of Sciences, Beijing 101408, China; 7Department of Medical Laboratory Sciences, College of Applied Medical Sciences, University of Bisha, P.O. Box 255, Bisha 67714, Saudi Arabia; mhahmad@ub.edu.sa; 8Department of Animal Nutrition and Nutritional Deficiency Diseases, Faculty of Veterinary Medicine, Mansoura University, Mansoura 33516, Egypt; shimaaradi_2009@mans.edu.eg; 9Department of Veterinary Medicine, College of Applied & Health Sciences, A’Sharqiyah University, P.O. Box 42, Ibra 400, Oman; karim.khalil@asu.edu.om; 10Group of Molecular Cell Biology and Bionanotechnology, Nanomedicine Department, Institute of Nanoscience and Nanotechnology, Kafrelsheikh University, Kafrelsheikh 33516, Egypt; nemany.hanafy@nano.kfs.edu.eg; 11Pathology Department, Faculty of Veterinary Medicine, Kafrelsheikh University, Kafrelsheikh 33516, Egypt; alaa_abdelaaty@vet.kfs.edu.eg

**Keywords:** chitosan, high-fat diet, apoptosis, oxidative stress, nrf2/kaep1, bcl-2/bax

## Abstract

**Simple Summary:**

Fatty liver injury is common in farmed fish, but its molecular mechanisms are not well understood. An eight-week feeding trial was conducted to investigate the effects of dietary chitosan on high-fat diet (HFD)-induced liver damage in Nile tilapia. Six diets with varying fat and chitosan levels were tested. Fish on the HFD showed increased growth, fat accumulation, elevated liver injury markers, and higher levels of pro-apoptotic and inflammatory markers. Chitosan supplementation mitigated these effects and reduced intestinal injury by improving antioxidant defense and reducing inflammation and apoptosis through the nrf2 and cox2 pathways.

**Abstract:**

Fatty liver injury is a prevalent condition in most farmed fish, yet the molecular mechanisms underpinning this pathology remain largely elusive. A comprehensive feeding trial spanning eight weeks was conducted to discern the potential of dietary chitosan in mitigating the deleterious effects of a high-fat diet (HFD) while concurrently exploring the underlying mechanism. Growth performance, haemato-biochemical capacity, antioxidant capacity, apoptotic/anti-apoptotic gene expression, inflammatory gene expression, and histopathological changes in the liver, kidney, and intestine were meticulously assessed in Nile tilapia. Six experimental diets were formulated with varying concentrations of chitosan. The first three groups were administered a diet comprising 6% fat with chitosan concentrations of 0%, 5%, and 10% and were designated as F6Ch0, F6Ch5, and F6Ch10, respectively. Conversely, the fourth, fifth, and sixth groups were fed a diet containing 12% fat with chitosan concentrations of 0%, 5%, and 10%, respectively, for 60 days and were termed F12Ch0, F12Ch5, and F12Ch10. The results showed that fish fed an HFD demonstrated enhanced growth rates and a significant accumulation of fat in the perivisceral tissue, accompanied by markedly elevated serum hepatic injury biomarkers and serum lipid levels, along with upregulation of pro-apoptotic and inflammatory markers. In stark contrast, the expression levels of nrf2, sod, gpx, and bcl-2 were notably decreased when compared with the control normal fat group. These observations were accompanied by marked diffuse hepatic steatosis, diffuse tubular damage, and shortened intestinal villi. Intriguingly, chitosan supplementation effectively mitigated the aforementioned findings and alleviated intestinal injury by upregulating the expression of tight junction-related genes. It could be concluded that dietary chitosan alleviates the adverse impacts of an HFD on the liver, kidney, and intestine by modulating the impaired antioxidant defense system, inflammation, and apoptosis through the variation in nrf2 and cox2 signaling pathways.

## 1. Introduction

The aquaculture sector has grown and prospered over the past few decades [1]. Despite its advancement, the aquaculture sector faces significant challenges, such as the limited availability of high-performing fry strains, the elevated cost of aquafeeds, and limited disease resistance [2].

In fact, short-term feeding of fish benefits from boosting dietary lipid levels within specified bounds [3]. However, as the concentration of dietary lipids increases, a series of adverse effects begins to emerge [4].

The primary organ in charge of regulating lipid uptake, synthesis, redistribution, catabolism, and storage is the liver. Steatosis is largely caused by fatty acid and triacylglycerol imbalances [5,6].

A high-fat diet is posited as the instigator of hepatic lipidosis in farmed fish [3,7,8,9]. This can subsequently compromise fish health and reduce production [10]. Fat accumulation has been shown in several studies utilizing animal models to trigger inflammation [11,12,13,14,15] and contribute to liver damage, hypoimmunity, and decreased appetite [8]. Furthermore, fat buildup can accelerate lipid peroxidation, inducing oxidative stress [16], and disrupt lipid metabolism, thereby inhibiting growth [3,8,10,17].

*nrf2* serves as a pivotal regulator of the adaptive response to oxidative stress. Upon exposure to oxidative stress, *nrf2* is activated through its dissociation from the kelch-like ECH-associated protein-1 (*Keap1*), subsequently translocating to the nucleus [18], where it activates a battery of antioxidant genes [19]. However, severe oxidative stress suppresses the *nrf2* pathway [20,21,22]. To counteract hepatic steatosis, *nrf2* activation inhibits lipogenesis and promotes beta-oxidation of fatty acids [23], while *nrf2* pathway inactivation could potentially exacerbate liver damage induced by hepatotoxicants [24,25]. Genetically regulated cell death is called apoptosis, and it may be brought on by a stream of physiological events. It is widely accepted in the scientific community that mitochondrial malfunction is a crucial indicator of apoptosis [26]. *cyt-c* is released by dysfunctional mitochondria, which then activates the downstream effector *cas-3*, resulting in the execution of apoptotic changes [27].

Occludin, a trans-membrane protein, members of the claudin family, junction adhesion molecules, and linker proteins such as *zo-1* are all components of tight junctions [28]. Increased intestinal permeability results from decreased intestinal tight junction protein expression, which can seriously jeopardize intestinal barrier integrity [29]. Impaired intestinal integrity is accompanied by intestinal permeability changes and histological changes [30,31,32]. The intestine predominantly responds to various stressors, including oxidative stress and inflammation [33,34]. Hence, oxidative stress and inflammation in the intestines are crucial for maintaining intestinal function. Consequently, to maintain intestinal function in the face of oxidative stress, an optimal candidate with free radical scavenging activity as well as antioxidant and anti-inflammatory capacity is urgently needed.

Natural materials have recently shown distinctive, affordable, and secure antibacterial properties. Recently, a remarkable number of organic bioactive substances have been investigated for potential use as methods of reducing the detrimental effects of an HFD in farmed fish [35,36,37].

Chitosan (β-(1-4)-N-acetyl-D-glucosamine) has proven to be among the most intelligent and safe natural cationic biopolymers. It is a member of a class of glycosaminoglycan-like polysaccharides having structural characteristics [38]. Following cellulose, chitosan is the second-most abundant polymer [39]. It offers exceptional qualities such as a non-toxic nature, biocompatibility, biodegradability, and increased solubility, along with immunological restoration attributes [40]. Chitosan is derived from the de-acetylation process of chitin and is recognized as a crucial constituent found in the exoskeletons of aquatic crustaceans, such as crab, crayfish, and shrimp, as well as in certain terrestrial organisms and the cell walls of some microorganisms [41]. Chitosan is receiving considerable interest and has already begun to contribute significantly to aquaculture’s sustainability. It satisfies environmental standards since it is an eco-friendly substance that also promotes the effective use of reagents and minimizes potential waste [42]. There are several uses for chitosan in the biomedical and pharmaceutical industries [15]. Chitosan is an active growth promoter that can be considered a crucial element for the growth of aquatic animals. The major effect of chitosan is improving the morphological structure of the small intestine, which may potentially augment nutrient absorption and subsequently improve growth performance [43]. Even at low concentrations, dietary chitosan was found to enhance nitrogen utilization and amino acid digestibility [44].

Nile tilapia (*Oreochromis niloticus*) is highly favored among cultured fish, ranking as the third most prominent group of cultured fish globally, surpassed by carp and salmonids. It is extensively cultured in about 100 countries in both tropical and subtropical regions [45,46]. The success of Nile tilapia in aquaculture is ascribed to its superior survival rate in poorly oxygenated environments, comparatively high resistance to diseases, and adaptability to consume a diverse range of foods [47]. Moreover, in cultured fish, there is a paucity of evidence supporting the potential pathogenesis of fatty liver injury. Consequently, a prevailing challenge in aquaculture is the development of strategies to mitigate the oxidative stress and inflammation induced by high-fat diets. Chitosan’s antioxidant activity has garnered significant attention. Research indicates that chitosan exhibits hepatoprotective effects due to its antioxidant properties [48]. Similarly, AlKandari et al. [49] stated that supplementing with natural antioxidants like propolis, chitosan, or their combination during ibuprofen use enhanced the reduction in toxic effects and improved the antioxidant system and anti-inflammatory response.

Our study constituted novel insights on hepatotoxicity and intestinal injury induced by an HFD and evaluated the ameliorative effects of dietary chitosan supplementation on intestinal integrity, oxidative status, and apoptosis/anti-apoptosis.

## 2. Materials and Methods

### 2.1. Ethical Approval

Following the normal operating procedures approved by the Institutional Animal Care and Animal Ethics Committee, Faculty Aquatic and Fisheries Sciences, Kafrelsheikh University, Egypt, the current experiment was conducted on Nile tilapia (*Oreochromis niloticus*) (IAACUC-KSU-028-2022).

### 2.2. Experimental Design

The experiment was carried out at the Fish Processing and Biotechnology department of the Faculty of Aquatic and Fisheries Sciences at Kafrelsheikh University in Egypt. A total of 216 juvenile Nile tilapia with an initial mean weight of 17.38 g ± 0.22 g (mean ± SD) were procured from a private farm located in Kafrelsheikh, Egypt. The fish were then acclimated to the experimental system for two weeks in glass aquariums and fed on the basal diet (Table 1) twice daily. Next, they were equally distributed among eighteen glass aquariums (80 × 45 × 35 cm, 12 fish/tank) with three replicates per treatment (six treatments, as shown in Table 2). Each aquarium was equipped with a mechanical filter (JAD, China) to eliminate waste from the water and an air stone for oxygen supply. Throughout the 60-day trial, water quality parameters, including temperature, pH, and dissolved oxygen, were monitored daily and recorded with an average of 26.19 ± 4 °C, 5.9 ± 0.8 mg/L, and 7.50 ± 0.1, respectively. The total ammonia level was checked once a week. All fish groups were fed twice daily with 4–6% of their body weight divided into two meals at 8:00 am and 3:00 pm.

### 2.3. Experimental Diet Designs

Six iso-nitrogenous (32% crude protein) test diets were designed as displayed in Table 1 and Table 2. The experimental diets were divided into two categories: normal fat diets (F6, 6% fat) and high-fat diets (F12, 12% fat), then each category was divided into three groups: chitosan-free diet (devoid of chitosan), low chitosan diet (5 g/kg diet), and high chitosan diet (10 g/kg diet) [50,51]. Diets were then extruded through a 2.33-mm-diameter die in a meat grinder, air-dried at room temperature, and stored in the freezer at −20 °C until used. A sample of each diet was collected and analyzed for proximate composition (g 100 g^−1^ as is) following AOAC (1995) procedures. Crude protein was determined based on the AOAC 990.03 Kjeldahl method by using the Kjeldahl apparatus (Infitek Co., Ltd. Shandong, China). Also, the Pet Ether Soxhlet Extraction Method (AOAC 945.16) was followed for the analysis of crude fat. Ash was measured using a muffle furnace at 600 °C for 5 to 6 h (AOAC 942.05). The digestible energy (DE), crude fiber (CF), calcium (Ca), and phosphorus (P) content of each diet were calculated using the DE, Ca, and P values stated in the feed ingredient composition table reported in NRC (1993).

### 2.4. Sample Collection

Following a fasting period of 24 h, all fish in each group were counted and weighed. From each aquarium tank, six fish (18 fish/group) were randomly chosen and anesthetized with FA-100 anesthetic (diluted 1:5000, DS Pharma Animal Health Company, Osaka, Japan). Of these, three fish per tank were designated for blood aspiration and tissue sampling (including liver, kidney, and intestine). Importantly, tissue samples were subsequently divided into two parts: the first part was kept in 10% formalin for histopathological analysis, while the other part was immediately submerged in liquid nitrogen and then stored at −80 °C for subsequent gene expression analysis. Furthermore, three fish from each tank (totaling nine fish per group) were randomly chosen for liver weighting and measurement of liver fat, hepato-somatic index (HSI), and intra-peritoneal fat index (IPF), and then the whole fish were stored at −20 °C to determine their chemical body composition.

### 2.5. Growth Performance and Morphometric Indices

The fish in each tank were weighed and counted at the beginning of the experiment, then biweekly, and at the end of the experiment. Growth and feed utilization parameters, including the initial body weight (IBW), final body weight (FBW), body weight gain (BWG), specific growth rate (SGR), feed conversion ratio (FCR), and protein efficiency rate (PER), were calculated. At the end of the experiment, 3 fish per tank were randomly chosen, euthanized with tricaine methane sulphonate, weighed, and dissected to obtain liver and visceral weights for further calculation of the hepato-somatic index (HIS) and viscerosomatic index (VSI). Also, from each group, 9 fish were analyzed (3 fish per tank) to determine the crude protein, crude fat, and ash content according to AOAC [52] guidelines. Liver samples were collected and analyzed to measure liver fat content according to the instructions of the Folch method [53].

The growth parameters were determined as follows:

Body weight gain (BWG, g) = FBW − IBW

Specific growth rate (SGR) (% body weight gain/day) = ((Ln FW − Ln IW)/t) × 100

Feed conversion rate (FCR) = Feed intake (g)/Weight gain (g)

Protein efficiency ratio (PER) = Live weight gain (g)/Dry protein intake (g)

Hepato-somatic index (HSI) = (Liver weight/Whole body weight) × 100

Intra-peritoneal fat (IPF) = (Visceral fat weight/Whole body weight) × 100

### 2.6. Hematological and Serum Biochemical Analysis

At the end of the experiment, two different blood samples were collected from the caudal vein using a 3 mL disposable syringe. The first blood sample was collected in tubes containing EDTA (Ethylenediaminetetraacetic acid) as an anticoagulant to quantify erythrocytes, hemoglobin (Hb), packed cell volume (PCV), total leukocyte counts, and differential leucocyte counts (monocytes, lymphocytes, heterophils, and eosinophils) according to [54]. The second blood sample was collected without anticoagulant and then centrifuged at 3000 rpm for 10 min at 4 °C using a benchtop refrigerated centrifuge (Heraeus Megafuge 8R, Thermo Fisher Scientific, Germany) to separate the serum, which has been stored at −20 °C until biochemical analyses. Commercial kits (BIODIGNOSTIC, Giza, Egypt) were used to test plasma for total proteins, albumin, cholesterol, triglycerides, high-density lipoprotein-C (HDL-C), low-density lipoprotein (LDL), urea, creatinine, glucose, globulins, aspartate aminotransferase (Ast), alanine aminotransferase (Alt), and lactate dehydrogenase (LDH). The globulin concentration (Glob) and consequently the albumin-to-globulin ratio (A/G) were calculated. VLDL-C concentrations were calculated using the standard Friedewald equation [55].

### 2.7. Histopathology Study

Liver specimens were immersed in 10% phosphate-buffered neutral formalin (dehydrated and cleared in xylene), and then all specimens were processed into paraffin blocks and cut off at a thickness of 5 μm. Sections were stained by standard histology techniques using hematoxylin and eosin, along with Masson trichrome for collagen fiber staining. Liver sections were oxidized with 1% periodic acid for 5 min, washed, and incubated in Schiff’s reagent for 10 min at room temperature. The sections were then washed and counterstained with Mayer’s hematoxylin for 30 s, washed, dehydrated, and mounted in DPX for histological analysis. Images were then acquired using an inverted light microscope at the Institute of Nanoscience and Nanotechnology, Kafrelsheikh University, Egypt (Leica Microsystems-Fluorescence Model, DMi8 manual, Wetzlar, Germany).

### 2.8. Total RNA Extraction, cDNA Synthesis, and RT-qPCR Assay

Total RNA was extracted from 50 mg of the livers and intestines of *Oreochromis niloticus* using GENEzolTM Reagent (Geneaid, UK) following the manufacturer’s instructions. The integrity of the extracted RNA was confirmed by electrophoresis using an ethidium bromide-stained 2% agarose gel. The concentration and purity of RNA were determined using a Nanodrop BioDrop spectrophotometer (Biochrom Ltd., Cambridge CB23 6DW, Cambridgeshire, UK) based on the A260/A280 nm ratio. Five μg of RNA samples were reverse transcribed using the TOP script TM RT Dry Mix (dt18/dN6 plus) kit (enzynomics, Daejeon, Republic of Korea). Gene expression profiling was performed in Rotor Gene-Q (Qiagen, 40724 Hilden, Germany) using gene-specific primer sequences for the amplification of immune-related genes such as *tumor necrosis factor alpha* (*tnfa*), *interleukin-1β* (*il1b*), *interleukin-10* (*il10*), and *cyclooxygenase 2* (*cox2*); antioxidant genes such as *nuclear factor-E2-related factor 2* (*nrf2*), *kelch-like ECH-associated protein 1* (*keap1*), *glutathione peroxidase* (*gpx*), and *superoxide dismutase* (*sod2*); and apoptosis genes such as *baxa*, *B-cell lymphoma 2* (*bcl2a*), *cytochrome c* (*cyc1*), *cysteine-aspartic acid protease 9* (*casp9*), and *cysteine-aspartic acid protease 3* (*casp3*). Tight junction genes, including *claudin 3* (*cldn3c*), *claudin 7* (*cldn7*), and *tight junction protein 1a* (*tjp1a*), encode for the Zonula occludens-1 protein, and *tight junction protein 1a* (*tjp2a*) encodes for the Zonula occludens-2 protein, as shown in Table 3 [56,57,58,59,60,61,62,63,64,65]. The amplification reaction was performed using the TOPrealTM qPCR 2X PreMIX (SYBR Green with Low Rox) kit (enzynomics, Daejeon, Republic of Korea). The reaction volume was adjusted to 20 μL consisting of 10 µL SYBR Green, 0.6 µL of forward and reverse specific primer, 1 µL of cDNA template, and nuclease-free water to make the final volume 20 µL. The PCR program was carried out with the following conditions: activation at 95 °C for 15 min, followed by 40 cycles of denaturation at 95 °C for 10 s, annealing at the primer-specific temperature for 15 s, and extension at 72 °C for 25 s. This was followed by a melt curve analysis to assess the specificity of amplification at 72 °C to 95 °C. The efficiency of amplification was estimated depending on the qPCR slope by using the formula: E = 10^−1/slope^. Normalization of qPCR data was performed using the geometric averaging of two internal reference genes: 18S ribosomal RNA (*18s rRNA*) and Ubiquitin C (*ubce*) to calculate fold change. All genes were tested in triplicate. CT values for each sample were determined and incorporated in an efficiency-corrected fold change (2^−ΔΔCT^) calculation based on [66], and mRNA expressions for each sample were normalized against the average of 18s ribosomal rRNA (*18srRNA*) and Ubiquitin C (*ubce*) as internal reference genes.

### 2.9. Statistical Analysis

Data are demonstrated as the mean ± standard error of the mean (M ± SEM). Preliminary, Shapiro–Wilk’s test was employed to assess data normality, while Levene’s test was used to evaluate variance homogeneity, both conducted with a significance level set at *p* < 0.05. Percentage data were subjected to arcsine transformation prior to analysis of variances. To investigate the differential effects of high fat and chitosan and their interaction, body indices, growth performances, hematological parameters, biochemical analyses, histo-morphometric measurements, and relative gene expression were analyzed using a two-way ANOVA, followed by Tukey’s multiple comparison test (*p* < 0.05). All statistical analyses were conducted using GraphPad Prism (version 9.5, GraphPad Software, San Diego, CA, USA).

## 3. Results

### 3.1. Effects on Growth, Feed Utilization, Morphometric Parameters, and Body Nutritional Composition

As presented in Table 4, no significant differences in final body weight (FBW), body weight gain (BWG), or specific growth rate (SGR) were detected among the experimental groups. Nile tilapia exposed to an HFD without chitosan supplementation (F12Ch0) showed a non-statistical decrease in BWG and SGR with the highest FCR, HSI, IPF, liver fat %, and body fat %, and the lowest PER and body protein % after 8 weeks of dietary intervention compared with the control untreated group and the other treatment groups. However, dietary supplementation with chitosan significantly reverses the aforementioned findings in a dose–response manner. Moreover, the ash content was significantly higher in F12Ch5 and F12Ch10 compared to the HFD-fed group (F12Ch0) and the other groups.

### 3.2. Hematological Parameters

As seen in Table 5, the erythrogram parameters showed significantly higher RBCs, Hb, and PCV in the F6Ch10 group compared to the control normal fat diet group and the other treatment groups. However, the lowest value for these parameters was detected in the HFD-fed group F12Ch0, while dietary chitosan supplementation significantly increased the RBC count in the F12Ch5 and F12Ch10 groups compared with the HFD-fed group F12C0. No significant differences were observed in mean corpuscular volume (MCV), mean corpuscular hemoglobin (MCH), and mean corpuscular hemoglobin concentration (MCHC) among all the groups. The leukogram parameter showed that the highest white blood cell (WBC) count, heterophil, lymphocytes, monocytes, and eosinophil counts were detected in the HFD-fed group (F12Ch0) compared with the control normal fat diet group and the other treatment groups. Dietary chitosan given to HFD-fed fish led to a significant decrease in the number of WBCs, lymphocytes, monocytes, and eosinophils, with a non-significant increase in heterophil count compared with the HFD-fed group (F12Ch0).

### 3.3. Serum Biochemical Parameters

As explained in Table 6, the results of serum biochemical findings exhibited significantly higher AST, ALT, and LDH enzyme activities and TG, TC, LDL-C, VLDL-C, creatinine, and glucose in the HFD-fed group (F12Ch0) compared with the control and the other treatment groups. There was no significant difference among the groups in blood urea nitrogen. There was a significant decrease in total proteins, albumin, globulins, and high-density lipoprotein (HDL) in the F12Ch0 and F12Ch5 groups compared to the control group and other treated groups. Moreover, chitosan supplementation either with or without an HFD succeeded in normalizing these parameters compared to the HFD-fed (F12Ch0) group in a dose–response manner.

### 3.4. Histopathological Examination

In Figure 1, the histopathological examination of kidney sections from different treatment groups revealed notable variations in renal architecture and cellular morphology. In the F6Ch0 group, the kidney displayed a histologically normal tubular and glomerular structure. The renal cells exhibited moderately vacuolar cytoplasm and were accompanied by adequate interstitial cells. In the F6Ch5 group, the histological structure was nearly normal, characterized by less vacuolated cytoplasm and only slight interstitial cell infiltration. However, the kidney sections from the F6Ch10 group showed focal tubular degeneration and interstitial inflammatory aggregations. Conversely, the F12Ch0 group exhibited severe renal damage, with diffuse tubular injury represented by severe necrotic tubular epithelial cells, intraluminal eosinophilic hyaline degeneration (renal casts), and interstitial fibrosis. These changes were accompanied by an infiltration of inflammatory cells and numerous RBCs, illustrating extensive renal damage in the high-fat diet group. In the F12Ch5 group, the kidney demonstrated mild tubular necrosis. Additionally, in the F12Ch10 group, the kidney sections showed mostly normal tubular endothelial lining with reduced cytoplasmic vacuolation and few interstitial inflammatory cell infiltrations.

On the other hand, the histopathological examination of intestinal sections from various treatment groups revealed distinctive findings in the intestinal architecture and mucosal integrity (Figure 2); however, no severe degenerative or necrotic changes were detected. In F6Ch0, the intestine exhibited a normal histological structure, characterized by well-preserved intestinal villi with an intact mucous membrane. In both the F6Ch5 and F6Ch10 chitosan groups, the intestinal mucosa remained mostly normal, with few intracellular vacuoles. Conversely, the F12Ch0 group displayed shortened intestinal villi with numerous vacuoles. In the F12Ch5 group, there was minimal apical mucosal loss, and the intestinal mucosa appeared expanded, reflecting partial improvement compared to the F12Ch0 group. Additionally, the F12Ch10 group showed elongated intestinal villi, slight apical mucosal loss, and a few intestinal vacuoles.

Regarding the histopathological examination of liver sections from different groups (Figure 3), the F6Ch0 group demonstrated a generally normal appearance, characterized by normal lobular architecture and normal hepatocytes with normal vacuoles of lipid and glycogen. In the F6Ch5 chitosan group, hepatocytes appeared normal, displaying normal lipid vacuoles with slight vascular dilatation. For the F6Ch10 chitosan group, signs of hydropic degeneration and slight blood vessel dilatation appeared, suggesting mild hepatocellular changes in response to chitosan supplementation. In contrast, the F12Ch0 group exhibited disarrangement of the lobular hepatic structure and marked hepatocellular necrosis, indicating severe liver damage; the F12Ch5 group showed a relatively normal hepatic structure with slight vascular dilatation; and the liver sections from the F12Ch10 group exhibited dilatation of the hepatic blood vessels with little inflammatory cell aggregation, hydropic degeneration of some hepatocytes, and increased eosinophilia of the cytoplasm, along with pyknosis of the nuclei.

Additionally, liver sections stained with Masson trichrome stain (Figure 4) showed the accumulation of collagen fibers in the HFD-fed group F12Ch0, while the control normal fat diet group showed an absence of collagen fibers. Groups F6Ch5 and F6Ch10, fed low and high chitosan doses for 8 weeks, respectively, displayed no collagen fibers. Nile tilapia fed with F12Ch5 had slight collagen fibers accumulated around blood vessels, while collagen fibers were completely reduced in those fed with F12Ch10.

Also, liver sections stained with periodic acid Schiff reagent (PAS) (Figure 5) in the control group revealed that glycogen granules occupied most of the cytoplasm of hepatocytes in F6Ch0. The HFD-fed group (F12Ch0) showed a reduction in PAS reaction compared to the control group. Glycogen granules appeared in groups of Nile tilapia fed with F6Ch5 and F6Ch10, respectively. Glycogen granules disappeared in Nile tilapia fed with F12Ch5, but initially appeared in those fed with F12Ch10.

At the same time, histomorphometric analysis of muscle tissue samples from all experimental groups showed the effect of adding chitosan to HFD-fed Nile tilapia (Table 7), where there was a significant improvement in muscle fiber count, total area, average size of muscle fiber, and area percent by adding chitosan to an HFD.

### 3.5. Differential Gene Expression Analysis

#### 3.5.1. Antioxidative Function-Related Gene Expression

Oxidative stress is considered a crucial factor in liver and intestinal injuries. We also examined the expression of critical genes involved in the antioxidant signaling pathway in the liver and intestine of tilapia after 60 days of HFD administration, as shown in Figure 6. The hepatic and intestinal *nrf2* expression levels exhibited a decreasing tendency (Figure 6A,B), but *kaep1* expression in the liver and intestine showed an upward tendency (Figure 6C,D) in HFD-treated tilapia (F12C0) compared with the control, and the other treated groups suggested that excess deposition of fat impaired the *nrf2* pathway and attenuated antioxidant defense. Moreover, reduced hepatic and intestinal expression levels of *gpx* (Figure 6E,F) and *sod* (Figure 6G,H) were also detected in HFD-treated tilapia (F12Ch0) compared with the control normal fat group and the other treated groups. However, dietary chitosan, either with or without an HFD, significantly reverses these findings in a dose-dependent manner.

#### 3.5.2. Apoptosis-Related Gene Expression

The expression levels of apoptosis-related genes are listed in Figure 7. The mRNA levels of *baxa* (Figure 7A), cytochrome c (*cytc*) (Figure 7C), caspase-3 (*cas-3*) (Figure 7D), and caspase-9 (Figure 7E) were greatly elevated. Conversely, the *bcl-2* inhibitor of apoptosis was signally downregulated in the liver and intestine of HFD-fed tilapia compared to the control and the other treated groups. However, dietary chitosan inclusion markedly downregulated the apoptotic genes while enhancing the anti-apoptotic gene (*bcl-2*) (Figure 7B).

#### 3.5.3. Tight Junction Gene Expression in the Intestine

As shown in Figure 7, an HFD significantly reduced the expression of tight junction protein genes in the intestine, including *claudin-3* (Figure 7G), *claudin-7* (Figure 7F), *tjp1* (Figure 7H), and *tjp2* (Figure 7I). Moreover, dietary chitosan significantly increased the levels of mRNA expression of *claudin-3*, *claudin-7*, *tjp1*, and *tjp2* compared with an HFD.

#### 3.5.4. Inflammatory-Related Gene Expression

Chronic inflammation was evaluated in fatty liver injury via measuring changes in *cox2* and other inflammatory factors. The mRNA levels of *cox2* and pro-inflammatory factors, including *tnf-α* and *il-1β*, were elevated by an HFD after 60 days of feeding (Figure 8). *tnf-α* (Figure 8C,D) and *il-1β* (Figure 8E,F) were markedly upregulated in the liver and intestine of the high-fat diet-fed group F12Ch0 in Nile tilapia after 60 days compared with F6Ch0 and the other treated groups. Moreover, hepatic and intestinal *il-10* expression levels were markedly inhibited compared with the control and the other treatment groups (Figure 8G,H). However, chitosan supplementation markedly reverses these observations compared with the HFD-fed group, F12Ch0.

## 4. Discussion

Lipids are a dietary energy resource that plays a consequential role in fish growth. However, excessive amounts of lipids in the diet will have adverse impacts on fish growth and health, including poor feed intake, slow growth, low immunity, and oxidative stress [67,68]. The formation of a fatty liver is mostly a result of the high content of fat in feed, which renders the body unable to consume too much fat itself. The deposition of fat will consequently cause an increase in the fat content in the fish body, which is principally deposited in liver cells and can result in a fatty liver [69].

In mammals, lipotoxicity is the primary contributor to several diseases associated with excess fat accumulation, such as a fatty liver, obesity, and diabetes [70,71]. Excessive deposition of triglycerides in the liver is considered an important biomarker of a fatty liver. Moreover, excessive dietary fat might also cause adverse effects on fish [72].

The negative impacts caused by an HFD in fish have been investigated in different species. It has been noticed in grass carp (*Ctenopharyngodon idella*) and tilapia (*Oreochromis niloticus*) that diets high in fat (15 or 16%) decreased feed intake and growth performance, lowered immunological function, and changed lipid metabolism [67,68,73]. Moreover, it has been reported in turbot (*Scophthalmus maximus*) and black seabream (*Acanthopagrus schlegelii*) that a high-fat diet (17–19.5%) induces lipid peroxidation and oxidative stress [9,74]. An HFD (11 or 12% fat) exacerbated intestinal and liver health in tilapia [17] and triggered endoplasmic reticulum (ER) stress to modulate hepatic lipid secretion in blunt snout bream (*Megalobrama amblycephala*) [3]. Excessive lipid consumption beyond the body’s capacity for utilization is the main risk factor for hepatic steatosis in most cultured fish. Hepatic steatosis is a typical form of abnormal lipid metabolism in the liver that is characterized by excessive deposition of triacylglycerol in hepatic tissue [3,72]. Growth, feed utilization rate, immunity, and stress tolerance are negatively impacted by steatosis [8,69,75]. It was explained by Tanaka et al. [76] that an HFD induces hepatic steatosis by inhibiting hepatic autophagic activity, which triggers fat accumulation. Similarly, in mammals, lipotoxicity is the main contributor to various diseases, such as a fatty liver, obesity, and diabetes [70,71]. The present study showed that Nile tilapia exposed to a high-fat diet without chitosan supplementation (F12Ch0) had the lowest values for FBW, BWG, SGR, PER, and body protein content and the highest values for FCR, HSI, IPF, liver, and body fat content after 8 weeks of a feeding trial. Tilapia groups offered trial diets fortified with 5 and 10 g/kg chitosan (100%) exhibited the opposite trend. Previous research studied the negative impacts of an HFD on tilapia and indicated that an HFD reduces feed intake, impairs growth performance, damages fish’s liver and intestinal tissue, and causes oxidative damage [75,77,78]. Also, Dai et al. [8] stated that an HFD-related inflammation response reduces the appetite of blunt snout bream and is closely related to lipid utilization abnormality and exacerbation of growth performance. The lowered growth performance and poor body indices obtained in Nile tilapia fed with an HFD (F12Ch0) could be due to the reduced feed intake, efficiency, digestion, absorption, and metabolism that resulted from the oxidative damage and inflammatory effect of an HFD on the liver and intestine, which are considered the main organs involved in nutrient digestion, absorption, and metabolism, as explained previously by Ding et al. [79]. This evidence is confirmed by the results of serum antioxidant enzymes and liver and intestine histopathology (Table 6 and Figure 2 and Figure 3). Subsequently, the nutrient deposition in the whole fish body, such as protein, decreased in this group, as seen in Table 4, as did the distorted tight junction proteins (Figure 8). Furthermore, excessive lipid intake causes over-deposition of fat in hepatic tissue and viscera, causing an increase in relative liver size (HSI), viscerosomal index (VSI), metabolic disturbances, and a reduction in liver activity [72,79,80,81]. Similar findings were observed in grass carp (*Ctenopharyngodon idella*) [13], turbot (Scophthalmus maximus) [74], black seabream (*Acanthopagrus schlegelii*) [9], largemouth bass (*Micropterus salmoides*), and orgiant croaker (*Nibea japonica*) fed with a high-fat diet [72,81].

In this feeding trial, Nile tilapia fed the high-fat diet (F12) supplemented with chitosan demonstrated better growth, feed utilization, body indices, body protein content, and lower fat accumulation in the body and liver. It indicates that adding chitosan mitigated the deleterious impacts induced by the high-fat diet (12% fat) without chitosan. The beneficial effects of chitosan supplementation on fish diets have been studied in different species. Chitosan plays various biological roles, including hypolipidemic, immunomodulatory, and antioxidant properties [82,83,84]. It has been observed that diets containing chitosan enhance the growth of shrimp *(Penaeus monodon)*, loaches *(Misgurnus anguillicadatus)*, and caspian kutum (*Rutilus frisii kutum Kamenskii*) fingerlings [85,86,87,88]. Similarly, Wu [50] noticed that diets containing 4 g kg^−1^ chitosan increased tilapia body weight gain, feed conversion rate, specific growth rate, body protein content, decreased lipid in the whole body, and hepatopancreas. It is linked to the modulatory action of chitosan on a variety of receptors, including the calcium-sensing receptor (CaSR), olfactory receptor, epidermal growth factor receptor, Tolllike receptor 4, TLR4/MD-2 receptor, scavenger receptor BI, and CYP7A1 [89,90]. Thus, certain receptors mediated the effects of chitosan, which in turn triggered the production of proteins. Furthermore, a large dose of chitosan probably displayed hypolipidemic action, restricting the tilapia from synthesizing lipids [83].

Hematological response is an essential index of fish health, which varies according to the type of stressor [91]. In the present study, it was notable that dietary chitosan positively impacts the hematological parameters in Nile tilapia by enhancing the RBC, PCV, and hemoglobin concentration. This may be attributed to the ability of chitosan to improve the absorption of macronutrients and micronutrients [92] that lead to good health. The count of WBCs is used to evaluate both fish innate immunity and health [93]. Moreover, chitosan-treated fish might receive a sufficient amount of oxygen for respiration and high metabolic activity [94,95]. Mubarak Ali et al. [96] stated that dietary chitosan and chitosan nanoparticles recorded better non-specific defenses due to the immunostimulatory effect and antimicrobial properties of chitosan nanoparticles. Similarly, Meshkini et al. [97] indicated that dietary chitosan at concentrations of 0.25% showed improving effects on the hematological indices, a result that agreed with the biochemical findings of the present study regarding enhanced total serum proteins in fish fed chitosan, which suggest better innate immunity of the fish. Comparable outcomes have been reported in many fish species when fed chitosan [98] and chitosan nanoparticles [99].

Fish biochemical parameters are regularly used as reliable diagnostic tools in biomonitoring, allowing for detection of the pathophysiological changes attributable to nutrition [100]. In this study, we found that fish fed an HFD, F12Ch0, showed significantly higher serum activities of AST, ALT, and LDH compared to the F6Ch0 group (Table 6). Their abnormal elevations implied the occurrence of both liver injury and hepatotoxicity, which are firmly associated with both hyperlipidemia and hepatic steatosis [101]. This is consistent with the work of Li et al. [102] and Chen et al. [103] in *M. amblycephala* and in blunt snout bream (*Megalobrama amblycephala*) fed an HFD. However, in the present study, AST and ALT activities were reduced by increasing dietary chitosan in HFD-fed fish, implying that dietary chitosan incorporation could mitigate the HFD-induced damage in Nile tilapia, which may be attributed to the antioxidative capability of chitosan that protects liver cells from damage. Current results coincided with Mehrpak et al. [104] and El-Naby et al. [92], who reported declining AST and ALT activities in fish fed a chitosan-supplemented diet. Elevated ALT and AST with raised creatinine levels may also affect other organs, such as the gills and kidney [105].

The concentrations of energetic metabolites triglycerides and cholesterol, as well as VLDL-C and LDL-C, were significantly increased in fish fed an HFD (F12Ch0). According to Mensinger et al. [106], cholesterol levels can result in disorders of lipid and lipoprotein metabolism, particularly liver dysfunction. Higher levels of lipid in fish feeds also make hepatocytes work harder, probably stressing them and contributing to liver damage [3]. Moreover, chitosan supplementation in an HFD-fed fish exhibited a beneficial effect on the lipid profile, as seen in a previous study on Nile tilapia (*Oreochromis niloticus*) [107,108]. Kang et al. [109] stated that chitosan oligosaccharide (COS) can inhibit the activity of pancreatic lipase and reduce the absorption of intestinal fat in combination with bile acids, as well as increase the excretion of fecal fat [110]. Accumulated shreds of evidence have demonstrated that these results were analogous to data in *B. bidyanus* [111], *C. carpio* [112], and *Ctenopharyngodon Idella* [113], and Xu et al. [114] indicated that adding 5% chitosan can promote liver LDL receptor (LDLR) mRNA expression, improve the clearance of LDL into the liver, and diminish the plasma cholesterol level. Furthermore, the current reduction in glucose levels may indicate the ability of chitosan to reduce liver gluconeogenesis and increase glucose consumption in the skeletal muscles [115].

Oxidative stress is a major cause of the progression of liver disease, which is induced by an HFD and leads to mitochondrial dysfunction, apoptosis, ER stress, and an inflammatory response [5,116]. Fish fed an HFD have a notable hepatic lipid accumulation and peroxidation [81,117]. As a result of lipid deposition and peroxidation, excessive production of ROS will deteriorate the integrity of organelles and cause severe damage to cells and tissues [81,118]. In this study, we found that the liver and intestine of HFD-fed Nile tilapia (F12Ch0) signally downregulated the mRNA levels of antioxidant enzymes (*SOD* and *GPX*), as well as *nrf2*, while elevating *kaep1*, which reflected the occurrence of severe oxidative stress and redox imbalance. *SOD*, *GSH-Px*, and *CAT* were considered the main antioxidant enzymes for free radical scavenging [119]. In this work, dietary chitosan could alleviate intestinal mucosal oxidative stress by upregulating the gene expression of the antioxidative enzyme activities of *SOD* and *GSH-PX*. These findings are consistent with Lan et al. [120], who indicated COS could increase *SOD*, *CAT*, *GSH-Px*, and *T-AOC* activity and decrease the *MDA* level after H2O2 challenge. Li et al. [121] also indicated COS enhanced *SOD* activity in the duodenum’s mucosa and decreased the *MDA* level in the jejunum and ileum’s mucosa in broilers. Furthermore, Li et al. [122] indicated that dietary COS supplementation elevated the inhibition of hydroxy radical capacity while decreasing *MDA* content in the ileum mucosa of broilers. On the same line, Assar et al. [123] stated that chitosan dietary inclusion in HFD-fed Zaraibi goat bucks caused a notable enhancement in antioxidant enzyme activities and suppressed the elevated MDA levels.

Many studies concluded that the accumulation of fat intensified vulnerability to oxidative stress and debilitated the antioxidant defense system in HFD-induced liver injury [116,124,125]. However, dietary chitosan enhanced nrf2 mRNA expression in the liver and intestine, similar to other similar studies on doxorubicin-challenged rats [126] and mice fed a high-fat diet [127].

Some studies have concluded that adding chitosan induces nuclear factor erythroid-derived2-like2 (*nrf2*) activation, which plays a vital role in cellular protection in opposition to free radical damage and reduces the incidence of severe oxidative stress as well as redox imbalance and chronic inflammation in the liver [3,8].

In this study, we explore the mechanism by which an HFD promotes hepatic and intestinal injury by focusing on mitochondrial dysfunction, which has been identified as a crucial driver for both cell injury and cell death. Apoptosis is a genetically programmed type of cell death that can be triggered by a variety of physiological changes. It is commonly accepted that mitochondrial dysfunction is a crucial indicator of apoptosis, which is mediated by the bcl-2 family of proteins, including both *bcl-2* and *bax* [26]. Mitochondrial dysfunction causes the release of *cyt c*, which activates the downstream effector *cas-3* and finally executes the apoptotic changes [27]. In the current investigation, apoptosis of hepatocytes was also associated with intrinsic mitochondrial pathways characterized by mitochondrial damage with the release of cytochrome c and activation of caspase-9 and, subsequently, caspase 3 as well as *baxa* in the HFD-fed group (F12Ch0). This is in line with Lu et al. [128], who detected that cytochrome c was released from mitochondria into the cytosol after feeding fish an HFD, causing damage to mitochondrial permeability. Cytochrome c release from mitochondria is considered the principal event for apoptosis. Therefore, we believe that the anti-apoptotic activity of chitosan stems from its ability to protect mitochondrial integrity. Among proapoptotic members, bax is perhaps the best-studied protein and is essential for mitochondrion-mediated apoptosis [129]. Functional analysis showed that bax could promote the release of cytochrome c from mitochondria [129,130]. Dietary chitosan supplementation significantly downregulated the expressions of cytochrome c, caspase-3, and caspase-9 compared to the HFD-fed group (F12Ch0). Conversely, *bcl-2* plays an anti-apoptosis role by ameliorating cytochrome c release, and a decrease in the *bax/bcl-2* ratio alleviates the amount of apoptosis [131,132]. In this study, dietary chitosan inclusion downregulated the bax expression level while upregulating the *bcl-2* expression level compared with the HFD-fed fish group, F12Ch0. An HFD induces hepatocyte apoptosis in the blunt snout bream [133]. In line with earlier findings, apoptosis was also enhanced through upregulating pro-apoptotic genes *cytc*, *bax*, *cas-3*, *cas-8*, and *p53* and downregulating anti-apoptotic genes *bcl-2* and *xiap* in HFD-fed tilapia [134]. Similarly, Dai et al. [8] observed an increase in hepatocyte gene expression of caspase-3, caspase-9, and CD68. These genes are known for their functions in promoting intrinsic apoptosis, regulating physiological cell death, and regulating pathological tissue degeneration in *M. amblycephala* fish.

High-fat diets are well recognized to induce metabolic inflammation throughout the tissues. High-fat diets elevate the amounts of endotoxins, circulating free fatty acids, and inflammatory mediators, resulting in low-grade systemic inflammation and disturbed homeostasis in many tissues [135]. Li et al. [122] indicated that long-term HFD feeding induced an increase in inflammatory markers in zebrafish (*Danio rerio*). Inflammatory cytokines are intimately associated with the development of metabolic disorders [131,132,136]. Many studies demonstrated that an HFD could impair lipid homeostasis and the rate of fat peroxidation and induce oxidative stress, thus inducing inflammatory responses in various marine fish species, notably black seabream (*Acanthopagrus schlegelii*), blunt snout bream (*Megalobrama amblycephala*), and large yellow croaker (*Larimichthys crocea*) [8,9,128,137,138]. We relate these adverse inflammatory responses to oxidative stress induced by an HFD diet (Figure 8). Previous investigations have revealed that oxidative stress precedes metabolic disorders induced by a high-fat diet in mice [139] and rats [140] and mitochondrial damage and apoptosis in rats [141]. The Nile tilapia, when under sustained oxidative stress, modulates the mRNA of *nrf2/kaep1* signaling as a signal for cell damage. Our results were consistent with earlier research demonstrating that inflammation was a key factor for liver injury after feeding an HFD to tilapia [142]. Meanwhile, dietary chitosan supplementation could attenuate inflammation by modulating *cox2* signaling molecules, increasing expression of anti-inflammatory markers *il-10*, and decreasing *tnf-α* and *il-1b* levels [124,143,144]. Bai et al. [145] reported that chitosan oligosaccharide supplementation resulted in downregulation of mRNA levels of pro-inflammatory cytokines (*il-6*, *tnf-α*, and monocyte chemo-attractant protein 1(*MCP-1*) in the liver of mice fed an HFD.

A healthy intestine morphology improves the capacity to absorb different nutrients and acts as an immune barrier [36]. However, consumption of an HFD causes an increase in intestinal permeability, impairs its mucosal defenses, and promotes intestinal inflammation [146]. A high-fat diet (15%) fed to Nile tilapia for a period of 8 weeks significantly reduced the length of intestinal villi, decreased the number of goblet cells, downregulated the mRNA expressions of tight junction proteins, i.e., *occludin* and *claudin*, and prompted the expression of the intestinal inflammatory factor *il-1b* [146]. The accumulating literature illustrates that inflammation is an important marker in intestinal dysfunction [147,148]. Chen et al. [147] demonstrated that overproduction of inflammatory cytokines could alter the intestinal permeability and tight junction structure by modulating tight junction-related gene expression in weaning piglets. At the same time, the overproduction of *il-1β*, *il-6*, and *tnf-α* directly causes intestinal mucosal injury [149]. Therefore, suppressing the overproduction of intestinal mucosal *il-1β*, *il-6*, and *tnf-α* proved to be effective in maintaining intestinal function. Previous research suggested that stressors could disturb the balance between anti-inflammatory and pro-inflammatory responses by boosting the production of pro-inflammatory cytokines [147,150]. In this study, intestinal *il-1b* and *tnf-α* levels were higher, whereas intestinal *il-10* levels were lower in the HFD-fed group compared to the F6Ch0 group, indicating that oxidative stress resulted in inflammation in the intestine. Dietary chitosan supplementation decreased the intestinal *il-1b* and *tnf-α* levels while enhancing the intestinal *il-10* levels, all of which was inconsistent with the results of Hu et al. [151], who reported that COS reduced *il-1β* and *tnf-α* mRNA expression levels in the jejunum mucosa of weaning pigs. Also, COS reduced the *il-6* and *tnf-α* mRNA expression levels in the liver of mice after being fed a high-fat diet [123]. These findings suggest that COS may mitigate intestinal inflammation by suppressing the levels of *il-1β*, *il-6*, and *tnf-α* [152,153].

Normal intestinal permeability is maintained by tight junctions, which are multi-protein complexes made up of various proteins, namely Zonula-Occludens, Occludin, and Claudin [154]. Paola et al. [155] suggested that intestinal hyperpermeability is a causative agent of both liver fibrosis and hepatic inflammation. A high-fat diet can significantly decrease the expression of tight junction proteins and thus increase intestinal permeability by weakening the intestinal barrier. Previous studies have demonstrated that mice fed with an HFD showed increased intestinal damage when compared to mice fed a low-fat diet [156].

In this study, HFD-induced oxidative stress downregulated the protein expression of *claudin3*, *claudin7*, *tjp1*, and *tjp2*, which reflected intestinal barrier dysfunction and was consistent with the results reported by Song et al. [157] and Cao et al. [158]. Moreover, dietary chitosan upregulated the expression of tight junction proteins, similar to other studies on mice fed high-fat diets [159], dexamethasone-challenged broilers [160], and weaning pigs [151,161]. These results indicated that chitosan could modulate oxidative-induced intestinal barrier function relatively well by maintaining the intestinal structure, intestinal permeability, and tight junction functionality.

In this investigation, we observed histological alterations in the liver, kidney, and intestine that are linked to a high-fat diet. These alterations were mainly in the form of renal tubular damage, hepatic necrosis, and shortening of intestinal villi. However, these changes showed less severe patterns emerging in the groups that received chitosan supplements.

Surprisingly, the F12Ch10 group had the best results, providing features of healing in the form of enlarged intestinal villi with a minor loss of apical mucosa, normal renal tubular endothelial lining, and restricted hepatic inflammatory cell infiltration, suggesting that increased chitosan supplementation may have a protective effect against fat droplet accumulation. Similar results were previously reported on common carp [162].

High fat storage in hepatocytes disrupts the normal regulation of glycogen metabolism in the liver. It decreases glycogen synthesis, increases gluconeogenesis, and impairs the response to insulin [163,164]. These changes, in turn, result in reduced glycogen storage in the liver and explain the low glycogen level in hepatocytes in the F12Ch0 group. Also, such results suggest the effect of chitosan treatment in alleviating fat accumulation and maintaining the normal metabolism of glucose, as observed by the higher storage of glycogen in hepatocytes from the F12Ch10 group.

Collectively, the histopathological findings suggest that chitosan supplementation, particularly at higher levels, may have a protective role in mitigating the severe damage induced by a high-fat diet.

In summary, the present study demonstrated that an HFD could promote liver and intestinal injury that might be closely related to oxidative damage, inflammation, and apoptosis. An impaired nrf2 pathway may depress the antioxidant defense system, leading to oxidative damage. Concurrently, apoptosis was induced by activating the mitochondria pathway after feeding an HFD, which triggered cox2, led to the release of pro-inflammatory factors, and exaggerated liver and intestinal injury.

## 5. Conclusions

This study suggests that dietary chitosan supplementation may be an effective nutritional strategy to mitigate the deleterious effects of HFD-induced oxidative stress via modulating the nrf2/kaep1 signaling pathway and anti-apoptotic action by increasing the expression of bcl-2. Therefore, it attenuates intestinal barrier damage by improving intestinal morphology and tight junction protein expression, which may be involved in suppressing intestinal inflammation and increasing antioxidant capacity in HFD-fed Nile tilapia.

## Figures and Tables

**Figure 1 biology-13-00486-f001:**
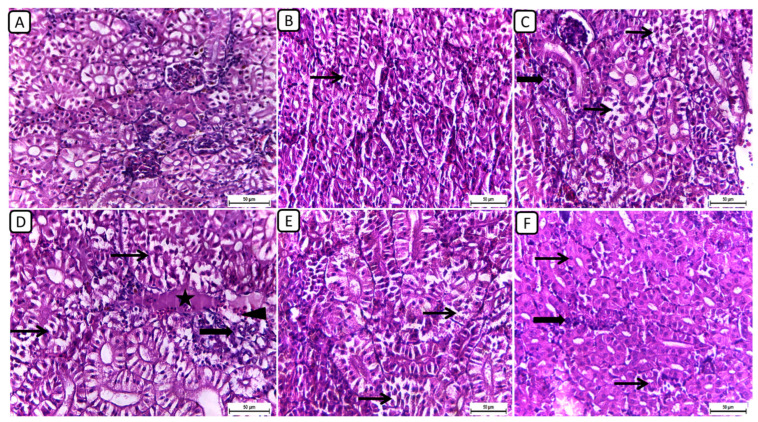
Representative photomicrographs of kidneys from different treatment groups. (**A**) F6Ch0 shows normal tubular and glomerular structure with moderately vacuolar cytoplasm in addition to adequate interstitial cells. (**B**) F6Ch5 shows nearly normal histological structure with less vacuolated cytoplasm in addition to slight interstitial cell infiltration (thin arrow). (**C**) F6Ch10 shows focal tubular degeneration (thin arrow) with interstitial inflammatory aggregations (thick arrow). (**D**) F12Ch0 shows diffuse tubular damage represented by severe necrotic tubular epithelial cells (thin arrows), intraluminal esinophilic cellular cast (star), and interstitial fibrosis (thick arrow) admixed with lymphocytes and numerous RBCs (arrowhead). (**E**) F12Ch5 shows mild tubular necrosis (thin arrows). (**F**) F12Ch10 shows mostly normal tubular endothelial lining with less vacuolation (thin arrows) and little interstitial inflammatory cell infiltration (thick arrow). Scale bar = 50 μm.

**Figure 2 biology-13-00486-f002:**
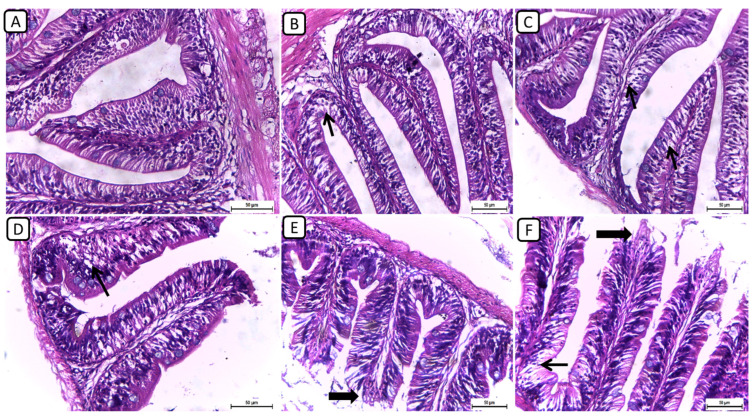
Representative photomicrographs of intestines from different treatment groups. (**A**) F6Ch0 shows a normal structure of intestinal villi with an intact mucous membrane. (**B**) F6Ch5 shows normal intestinal mucosa except for a few intestinal vacuoles (thin arrow). (**C**) F6Ch10 shows mild intestinal vacuolation (thin arrows). (**D**) F12Ch0 shows shortened intestinal villi with many vacuoles (thin arrow). (**E**) F123Ch5 shows minimal apical loss (thick arrow) and expanded intestinal mucosa. (**F**) F12Ch10 shows elongation of the intestinal villi, slight apical mucosal loss (thick arrow), and little intestinal vacuolation (thin arrow). Scale bar = 50 μm.

**Figure 3 biology-13-00486-f003:**
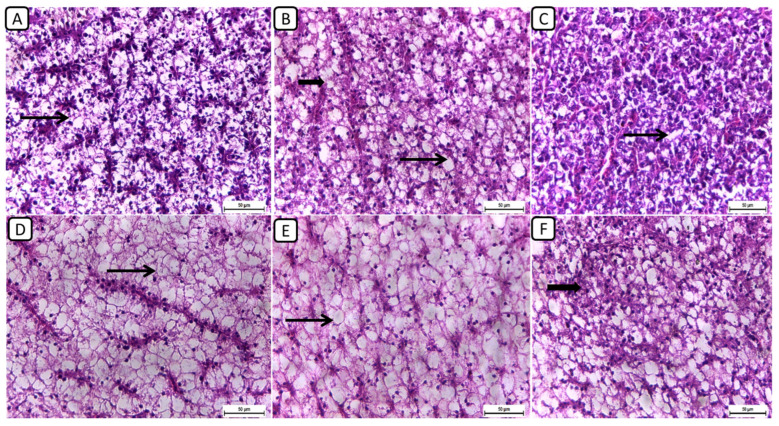
Representative photomicrographs of livers from different treatment groups. (**A**) F6Ch0 shows normal hepatic architecture with a normal vacuolation appearance. (**B**) F6Ch5 shows mostly normal hepatocytes and normal lipid vacuoles (thin arrow) with a few slight degenerative changes (thick arrow). (**C**) F6Ch10 shows hydropic degeneration (thin arrow) with slight blood vessel dilatation. (**D**) F12Ch0 shows marked hepatocellular necrosis with a loss of most nuclei (thin arrow). (**E**) F12Ch5 shows relatively vacuolated cytoplasm with intact nuclei. (**F**) F12Ch10 shows normal hepatocytes with slight degenerative changes in some cells (thick arrow). Scale bar = 50 μm.

**Figure 4 biology-13-00486-f004:**
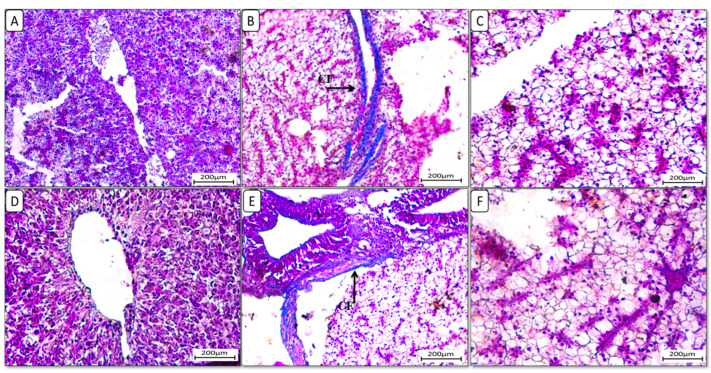
Liver histological morphology in Nile tilapia fed with different experimental diets for 8 weeks. Liver tissue sections were stained with Masson trichrome stain. (**A**): F6Ch0; (**B**): F12Ch0 (accumulation of collagen fibers (CFs)); (**C**): F6Ch5; (**D**): F6Ch10; (**E**): F12Ch5 (mild accumulation of collagen fibers (CFs)); (**F**): F12Ch10. Scale bar = 200 μm.

**Figure 5 biology-13-00486-f005:**
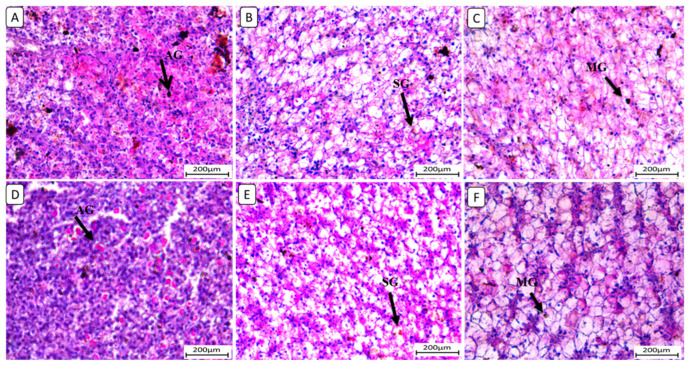
Histological liver sections of Nile tilapia fed with different experimental diets for 8 weeks. Tissue sections were stained with a periodic acid Schiff reagent (PAS) stain. (**A**): F6Ch0 (accumulated glycogen (AG)); (**B**): F12Ch0 (slight glycogen (SG)); (**C**): F6Ch5 (mild glycogen (MG)); (**D**): F6Ch10 (accumulated glycogen (AG)); (**E**): F12Ch5 (slight glycogen (SG)); (**F**): F12Ch10 (mild glycogen (MG)). Scale bar = 200 μm.

**Figure 6 biology-13-00486-f006:**
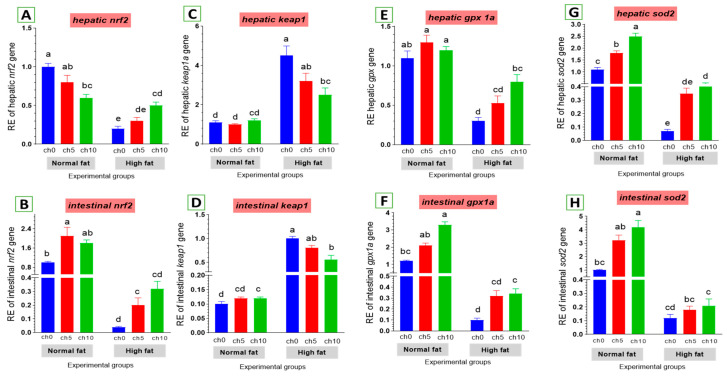
Differential expression of different antioxidant genes in the liver and intestine of Nile tilapia groups fed on normal and high-fat diets with chitosan. (**A**,**B**) Nrf2: Nuclear factor erythroid 2-related factor 2, (**C**,**D**) Kaep1: Kelch-like ECH-associated protein 1, (**E**,**F**) GPx: Glutathione peroxidase, (**G**,**H**) SOD: Superoxide dismutase. Columns with different superscript letters in the same figure are significantly different (*p* ≤ 0.05).

**Figure 7 biology-13-00486-f007:**
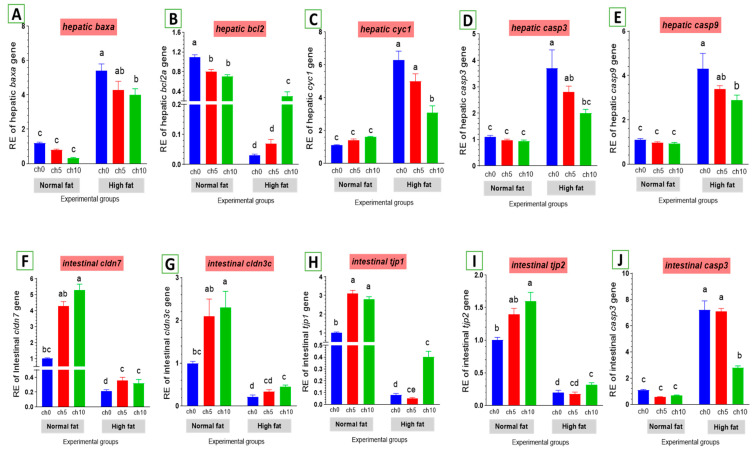
Differential expression of apoptosis-related genes in the liver and tight junction-related genes in the intestine of Nile tilapia groups fed on normal and high-fat diets with chitosan. (**A**) Bax: Bcl-2 associated X-protein, (**B**) Bcl2: B-cell lymphoma 2, (**C**) Cyc1: Cytochrome c, (**D**) Casp3: Cysteine-aspartic acid protease3 in the liver, (**E**) Casp9: Cysteine-aspartic acid protease9, (**F**) cldn7: Claudin7, (**G**) cldn3c: Claudin3c, (**H**) tjp1: Zonula occludens-1, (**I**) tjp2: Zonula occludens-2, (**J**) casp3: Cysteine-aspartic acid protease3 in the intestine. Columns with different superscript letters in the same figure are significantly different (*p* ≤ 0.05).

**Figure 8 biology-13-00486-f008:**
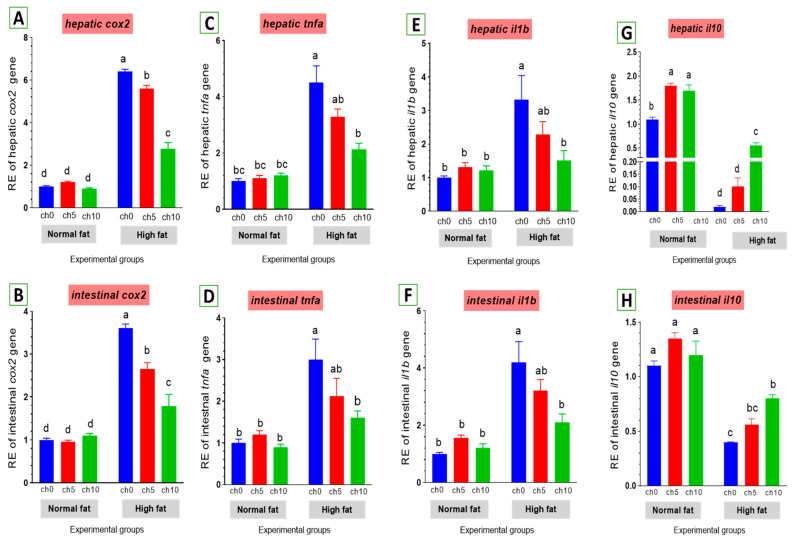
Differential expression of immune-related genes in the liver and intestine of Nile tilapia groups fed on normal and high-fat diets with chitosan. (**A**,**B**) *cox2*: Cyclooxygenase 2 in the liver and intestine, (**C**,**D**) *tnf-a*: Tumor necrosis factor alpha, (**E**,**F**) *il-1β*: Interleukin-1beta, (**G**,**H**) *il-10*: Interleukin-10. Columns with different superscript letters in the same figure are significantly different (*p* ≤ 0.05).

**Table 1 biology-13-00486-t001:** Ingredient and proximate compositions (g/kg, as-fed) of the normal and high-fat diets supplemented with low and high doses of chitosan at both levels of dietary fat.

	F6Ch0	F6Ch5	F6Ch10	F12Ch0	F12Ch5	F12Ch10
Ingredients (g/kg)						
Fish meal (65%)	100	100	100	100	100	100
Soybean meal (45%)	463	463	463	463	463	463
Corn gluten meal	30	30	30	30	30	30
Yellow corn	164	165	170	135	125	115
Wheat flour	186	180	170	152	157	162
Soybean oil	35	35	35	98	98	98
Vitamin mixture *	8	8	8	8	8	8
Mineral mixture **	5	5	5	5	5	5
DiCaP	6	6	6	6	6	6
Choline chloride	2	2	2	2	2	2
Stay C ***	1	1	1	1	1	1
Chitosan (100%)	0	5	10	0	5	10
Composition (%)						
Crude protein	32.20	32.14	32.07	31.55	31.53	31.51
DE (Kcal/Kg)	3004.63	2998.17	2997.59	3365.90	3343.24	3320.58
Crude lipid	5.84	5.83	5.84	11.99	11.96	11.94
Ash	5.167	5.166	5.168	5.116	5.10	5.09
Crude fiber	4.11	4.10	4.10	4.00	3.98	3.97
Ca	0.798	0.798	0.797	0.796	0.796	0.796
P	0.824	0.823	0.821	0.804	0.803	0.802

* Vitamin (g/kg premix): Thiamin HCl, 0.44; Riboflavin, 0.63; Pyridoxine HCl, 0.91; DL pantothenic acid, 1.72; Nicotinic acid, 4.58; Biotin, 0.21; Folic acid, 0.55; Inositol, 21.05; Menadione sodium bisulfite, 0.89; Vitamin A acetate, 0.68; Vitamin D3, 0.12; dL-alpha-tocoperol acetate, 12.63; Alpha-cellulose, 955.59. ** Trace mineral (g/100 g premix): Cobalt chloride, 0.004; Cupric sulfate pentahydrate, 0.25; Furrous sulfate, 4.000; Magnesium sulfate anhydrous, 13.862; Manganous sulfate monohydrate, 0.650; Potassium iodide, 0.067; Sodium selenite, 0.010; Zinc sulfate hepahydrate, 13.193; Alpha-cellulose, 67.964. *** Stay C^®^ (L-ascorbyl-2-polyphosphate, 35%).

**Table 2 biology-13-00486-t002:** Experimental groups.

Group Code	Treatments
F6Ch0	Normal fat diet (6% fat and 0% chitosan)
F6Ch5	Normal fat diet + low chitosan (5 g/kg)
F6Ch10	Normal fat diet + high chitosan (10 g/kg)
F12Ch0	High-fat diet 12% (By adding 63 g of soybean oil to 1 kg of a normal fat diet)
F12Ch5	High-fat diet + low chitosan (5 g/kg)
F12Ch10	High-fat diet + high chitosan (10 g/kg)

**Table 3 biology-13-00486-t003:** Primers used for qRT-PCR analysis.

Gene	Primer Sequence (5′-3′)	NCBI Gene Bank Accession No.	Reference
Internal reference genes			
*18srRNA*	F: GGACACGGAAAGGATTGACAG	JF698683	[56]
	R: GTTCGTTATCGGAATTAACCAGA		
*ubce*	F: CTCTCAAATCAATGCCACTTCC	XM_003460024	
	R: CCCTGGTGGAGGTTCCTTGT		
Inflammation-related genes			
*tnf-α*	F: GGAAGCAGCTCCACTCTGATGA	JF957373.1	[57]
	R: CACAGCGTGTCTCCTTCGTTCA		
*il1β*	F:CAAGGATGACGACAAGCCAACC	XM_003460625.2	
	R:AGCGGACAGACATGAGAGTGC		
*il10*	F:CTGCTAGATCAGTCCGTCGAA	XM_003441366.2	[58]
	R: GCAGAACCGTGTCCAGGTAA		
*cox2*	F:AGCAGCCAGAAGGAAGGCGG	-	[59]
	R:GACTGAGTTGCAGTTCTCTTAGTGTGC		
oxidative stress response genes			
*nrf2*	F: CTGCCGTAAACGCAAGATGG	XM_003447296.4	[60]
	R: ATCCGTTGACTGCTGAAGGG		
*keap1*	F: CTTCGCCATCATGAACGAGC	XM_003447926.3	
	R: CACCAACTCCATACCGCACT		
*gpx 1a*	F: TCGGACATCAGGAGAACTGC	GQ853451.1	[61]
	R: GCACTGCTCAAAGTTCCAGG		
*sod2*	F: CATGCCTTCGGAGACAACAC	AY491056.1	[61]
	R: ACCTTCTCGTGGATCACCAT		
Apoptosis-related genes			
*baxa*	F: GAGCAAGGTGGCTGGGAGG	MH370850.1	[62]
	R:TGCGAATGACAAGAACAGTGGTAAG		
*bcl2a*	F:ACGCAGGCATCCACAGAGTC	XM_003437902.5	[62]
	R:TCTATCACCTCGGCGAACCTC		
*cyt-c1*	F: GCTGAGCCGGTTACTTACCT	XM_005473699.4	This study
	R: CTGCTTGTCCGGTCTTCCTT		
*casp9*	F: CTTCAGCGGAACAGGGTTA	XM_025901776.1	[63]
	R:GAAGGCACTCCAGAAATAAGG		
*casp3*	F: GGCTCTTCGTCTGCTTCTGT	GQ421464.1	[58]
	R:GGGAAATCGAGGCGGTATCT		
Genes for Tight junction			
*cldn3c*	F:GCAACATTGTGACGGCTCAGAT	XM_005465025.3	[64]
	R:AGAGGGCGAGCATAGAGTCATACA		
*cldn7*	F:TGGCAGCAACATAGCAAAGG	XM_019347969.1	[65]
	R:GATGACGAGAATGGCAGATGC		
*tjp1*	F:ACAGGGTGTGAAGAACATGAGGAC	XM_013270540.2	[64]
	R:AATGGCTCGCTCATAGAGCTTCC		
*tjp2*	F:GCTTTGGCATTGCTGTATCAG	XM_019361305.1	[65]
	R:AACGAGTGGATGGCTCCATC		

Internal reference genes (*18srRNA*: 18s ribosomal rRNA, *UBCE*: Ubiquitin C), *tnfα*: Tumor necrosis factor alpha, *il-1β*: Interleukin-1beta, *il-10*: Interleukin-10, cox2: Cyclooxygenase 2, *nrf2*: Nuclear factor -E2-related factor 2, *keap1*: Kelch-like ECH-associated protein 1, *gpx*: Glutathione peroxidase, *sod2*: Superoxide dismutase, *baxa*: bcl2 Associated X, Apoptosis Regulator *bcl2a*: B-cell lymphoma 2, *cyc1*: Cytochrome c, *casp9*: Cysteine-aspartic acid protease9, *casp3*: Cysteine-aspartic acid protease3, *cldn3c*: Claudin3, *cldn7*: Claudin7, *tjp1*: Zonula occludens-1, *tjp2*: Zonula occludens-2.

**Table 4 biology-13-00486-t004:** Growth, feed utilization, morphometric indices, and proximate analysis of Nile tilapia (*Oreochromis niloticus*) fed on normal and high-fat diets supplemented with low and high doses of chitosan at both levels of dietary fat for 8 weeks.

Parameters	F6C0	F6C5	F6C10	F12C0	F12C5	F12C10	*p* Value of Two-Way ANOVA		
							Fat	Chitosan	Interaction
IBW (g)	17.42 ± 0.63	17.42 ± 0.50	17.29 ± 0.61	17.13 ± 0.49	17.92 ± 0.59	17.50 ± 0.56			
FBW (g)	35.80 ± 0.80 ^ab^	33.12 ± 0.94 ^ab^	35.47 ± 1.03 ^ab^	32.29 ± 0.70 ^b^	34.24 ± 0.98 ^ab^	37.17 ± 1.12 ^a^	0.7670	0.0199	0.0201
BWG (g)	18.38 ± 0.94 ^ab^	15.70 ± 0.69 ^b^	18.18 ± 0.87 ^ab^	15.17 ± 0.85 ^b^	16.32 ± 0.75 ^ab^	19.67 ± 1.07 ^a^	0.6107	0.0076	0.0289
Total feed intake (g)	38.2 ± 0.09 ^ab^	36.7 ± 0.45 ^bc^	37 ± 0.45 ^bc^	36.3 ± 0.18 ^c^	38.5 ± 0.72 ^ab^	39.1 ± 0.27 ^a^	0.0597	0.1737	<0.0001
SGR (% day^−1^)	3.52 ± 0.30	3.44 ± 0.21	3.51 ± 0.19	3.42 ± 0.21	3.48 ± 0.25	3.56 ± 0.19	0.9857	0.9372	0.9335
FCR	2.08 ± 0.06 ^bc^	2.34 ± 0.08 ^ab^	2.04 ± 0.08 ^c^	2.41 ± 0.05 ^a^	2.36 ± 0.06 ^ab^	2 ± 0.05 ^c^	0.0619	0.0001	0.0187
PER	1.60 ± 0.07 ^ab^	1.51 ± 0.04 ^bc^	1.59 ± 0.05 ^ab^	1.33 ± 0.05 ^c^	1.45 ± 0.06 ^bc^	1.82 ± 0.07 ^a^	0.4895	0.0005	0.0010
Survival (%)	95.83 ± 5.9	91.667 ± 11.8	100 ± 0	91.66 ± 11.8	83.33 ± 11.8	83.33 ± 11.8	0.0130	0.3725	0.3725
HSI (%)	2.85 ± 0.14 ^a^	2.38 ± 0.13 ^ab^	1.72 ± 0.13 ^b^	3 ± 0.17 ^a^	2.75 ± 0.15 ^a^	2.97 ± 0.20 ^a^	<0.0001	0.0034	0.0035
IPF (%)	0.53 ± 0.05 ^c^	0.19 ± 0.05 ^d^	0.17 ± 0.04 ^d^	1.19 ± 0.06 ^b^	1.46 ± 0.08 ^a^	1.08 ± 0.04 ^b^	<0.0001	0.0006	<0.0001
Liver fat (%)	14.29 ± 0.62 ^b^	13.33 ± 0.59 ^b^	12.50 ± 0.65 ^b^	20 ± 0.85 ^a^	17.62 ± 0.50 ^a^	14.71 ± 0.62 ^b^	<0.0001	<0.0001	0.0397
Body fat (%)	29.75 ± 0.36 ^a^	25.70 ± 0.74 ^ab^	23.47 ± 0.47 ^b^	28.28 ± 0.40 ^a^	21.25 ± 0.39 ^a^	27.61 ± 0.49 ^a^	0.1073	<0.0001	<0.0001
Body protein (%)	52 ± 0.58 ^ab^	54.49 ± 0.59 ^a^	52.34 ± 0.65 ^ab^	48.44 ± 0.49 ^c^	51.29 ± 0.75 ^b^	52.15 ± 0.71 ^ab^	0.0002	0.0008	0.0259
Body ash (%)	10.48 ± 0.43 ^d^	13.70 ± 0.55 ^bc^	12.77 ± 0.50 ^cd^	13.50 ± 0.49 ^bc^	16.47 ± 0.73 ^a^	16.01 ± 0.75 ^ab^	<0.0001	<0.0001	0.9236

IBW: Initial body weight, FBW: Final body weight, BWG: Body weight gain, SGR: Specific growth rate, FCR: Feed conversion ratio, PER: Protein efficiency rate, HSI: Hepatosomatic index, IPF: Intraperitoneal fat index. Data are expressed as the mean ± SEM, where *n* = 3 as triplicate tanks for BWG%, FCR, SGR, PER, HSI, IPF, liver fat, body fat, body protein, and body ash, and *n* = 45 for IBW and FBW. Values, or groups, with different superscript letters within a row are significantly different (*p* < 0.05) from each other based on the results of the two-way ANOVA and post-hoc multiple comparison tests.

**Table 5 biology-13-00486-t005:** Hematological parameters of Nile tilapia (*Oreochromis niloticus*) fed on normal and high-fat diets supplemented with low and high doses of chitosan at both levels of dietary fat for 8 weeks.

Parameters	F6Ch0	F6Ch5	F6Ch10	F12Ch0	F12Ch5	F12Ch10	*p* Value of Two-Way ANOVA		
							Fat	Chitosan	Interaction
RBCs (×10^6^/µL)	2.87 ± 0.03 ^b^	2.90 ± 0.06 ^b^	3.43 ± 0.04 ^a^	2.60 ± 0.03 ^c^	2.79 ± 0.08 ^bc^	2.95 ± 0.03 ^b^	<0.0001	< 0.0001	0.0049
Hb (g/dL)	8.71 ± 0.04 ^c^	9.51 ± 0.11 ^b^	10.39 ± 0.09 ^a^	8.19 ± 0.08 ^d^	8.53 ± 0.02 ^cd^	9.27 ± 0.11 ^b^	<0.0001	< 0.0001	0.0035
PCV (%)	28 ± 0.63 ^bc^	31 ± 0.4 ^a^	33.67 ± 0.26 ^a^	24.67 ± 2.62 ^c^	28 ± 0.77 ^bc^	29.67 ± 0.52 ^ab^	0.0016	0.0006	0.9119
MCV (fL)	97.70 ± 1.02	107.02 ± 3.69	98.45 ± 0.52	95.03 ± 10.51	100.53 ± 1.53	100.62 ± 2.3	0.5497	0.3059	0.6591
MCH (pg)	30.40 ± 0.21	32.82 ± 1.06	30.38 ± 0.12	31.54 ± 0.65	30.68 ± 0.86	31.42 ± 0.39	0.9861	0.3638	0.0290
MCHC (g/dL)	31.14 ± 0.54	30.67 ± 0.18	30.86 ± 0.05	34.70 ± 4.23	30.53 ± 0.88	31.25 ± 0.39	0.3919	0.4029	0.5423
WBCs (×10^3^/μL)	20.80 ± 0.41 ^b^	18.33 ± 3.86 ^b^	22.84 ± 1.60 ^ab^	24.80 ± 1.39 ^ab^	29.15 ± 0.79 ^a^	20.93 ± 0.71 ^b^	0.0091	0.6131	0.0083
Heterophils (×10^3^/μL)	3.22 ± 0.001 ^a^	2.25 ± 0.14 ^b^	2.85 ± 0.36 ^ab^	3.15 ± 0.22 ^a^	2.63 ± 0.20 ^ab^	2.24 ± 0.2 ^b^	0.5482	0.0026	0.0697
Lymphocytes (×10^3^/μL)	15.71 ± 0.51 ^b^	14.06 ± 3.44 ^b^	17.60 ± 0.92 ^ab^	19.32 ± 0.98 ^ab^	23.80 ± 0.6 ^a^	16.80 ± 0.46 ^b^	0.0030	0.5040	0.0087
Monocytes (×10^3^/μL)	1.46 ± 0.03	1.56 ± 0.41	1.94 ± 0.3	1.90 ± 0.16	2.23 ± 0.08	1.47 ± 0.07	0.1004	0.6110	0.1454
Eosinophils (×10^3^/μL)	0.21 ± 0.003	0.23 ± 0.1	0.16 ± 0.06	0.17 ± 0.07	0.29 ± 0.01	0.14 ± 0.06	0.9539	0.1870	0.6884
Basophils (×10^3^/μL)	0.20 ± 0.13	0.23 ± 0.1	0.30 ± 0.05	0.25 ± 0.01	0.19 ± 0.07	0.14 ± 0.06	0.4423	0.9818	0.4449

RBCs: Red blood corpuscles, Hb: Hemoglobin, PCV: Packed cell volume, MCV: Mean cell volume, MCH: Mean cell hemoglobin, WBCs: White blood cells. Data are expressed as the mean ± SEM, where *n* = 5. Parameters, values, or groups with different superscript letters within a row are significantly different (*p* < 0.05) from each other based on the results of the two-way ANOVA and post-hoc multiple comparison tests.

**Table 6 biology-13-00486-t006:** Biochemical parameters of Nile tilapia (*Oreochromis niloticus*) reared for 8 weeks and fed on normal and high-fat diets supplemented with low and high doses of chitosan at both levels of dietary fat.

Parameters	F6Ch0	F6Ch5	F6Ch10	F12Ch0	F12Ch5	F12Ch10	*p* Value of Two-Way ANOVA		
							Fat	Chitosan	Interaction
Total protein (g/dL)	4.90 ± 0.08 ^a^	5.17 ± 0.05 ^a^	5.37 ± 0.10 ^a^	3.77 ± 0.23 ^b^	4.10 ± 0.09 ^b^	4.87 ± 0.11 ^a^	<0.0001	<0.0001	0.0336
Albumin (g/dL)	1.24 ± 0.02 ^ab^	1.24 ± 0.10 ^ab^	1.41 ± 0.03 ^a^	0.73 ± 0.18 ^c^	0.95 ± 0.05 ^bc^	1.22 ± 0.015 ^ab^	0.0001	0.0037	0.2118
Globulins (g/dL)	3.66 ± 0.1 ^a^	3.93 ± 0.15 ^a^	3.96 ± 0.11 ^a^	3.03 ± 0.05 ^c^	3.15 ± 0.14 ^bc^	3.65 ± 0.10 ^ab^	<0.0001	0.0021	0.1408
A/G	0.34 ± 0.015	0.32 ± 0.036	0.36 ± 0.015	0.27 ± 1.65	0.30 ± 0.03	0.33 ± 0.01	0.0688	0.1775	0.4708
HDL (mg/dL)	48.33 ± 2.58 ^ab^	50.66 ± 2.91 ^ab^	54.66 ± 2.58 ^a^	31.33 ± 0.54 ^c^	38.33 ± 1.29 ^bc^	47.66 ± 3.36 ^ab^	0.0038	<0.0001	0.2743
VLDL (mg/dL)	28.80 ± 1.41 ^b^	26.20 ± 1.12 ^b^	24.47 ± 0.72 ^b^	41.33 ± 1.63 ^a^	36.27 ± 1.84 ^a^	29.1 ± 1.78 ^b^	<0.0001	< 0.0001	0.0382
LDL (mg/dL)	97.87 ± 9.73 ^b^	74.80 ± 6.08 ^bc^	53.20 ± 0.96 ^c^	205 ± 7.47 ^a^	141 ± 11.82 ^a^	77 ± 10.70 ^bc^	<0.0001	< 0.0001	0.0006
Cholesterol (mg/dL)	175 ± 8.50 ^c^	151.67 ± 5.59 ^cd^	132.33 ± 8.96 ^d^	277.67 ± 9.64 ^a^	216 ± 13.69 ^b^	145.67 ± 7.83 ^cd^	<0.0001	< 0.0001	0.0011
Triglycerides (mg/dL)	144 ± 7.03 ^b^	131 ± 5.60 ^b^	122.33 ± 3.61 ^b^	206.67 ± 8.13 ^a^	181.33 ± 9.21 ^a^	145.67 ± 8.88 ^b^	<0.0001	<0.0001	0.0382
Glucose (mg/dL)	42.33 ± 2.73 ^bc^	36 ± 2.05 ^c^	29 ± 2.93 ^c^	65.67 ± 2.91 ^a^	57 ± 2.72 ^ab^	42.33 ± 0.74 ^bc^	<0.0001	<0.0001	0.3154
Creatinine (mg/dL)	0.45 ± 0.004 ^c^	0.43 ± 0.002 ^cd^	0.36 ± 0.02 ^d^	0.96 ± 0.015 ^a^	0.57 ± 0.02 ^b^	0.56 ± 0.02 ^bc^	<0.0001	<0.0001	<0.0001
Blood Urea Nitrogen (mg/dL)	8 ± 1.61	7.38 ± 0.47	6.33 ± 0.26	9.67 ± 1.01	8.83 ± 0.68	8.03 ± 0.15	0.0480	0.1633	0.8994
Ast (U/L)	35.13 ± 0.46 ^b^	22.50 ^cd^	15.67 ± 0.26 ^d^	61.67 ± 4.64 ^a^	35.50 ± 2.53 ^b^	26.83 ± 2.03 ^bc^	<0.0001	<0.0001	0.0054
Alt (U/L)	35 ± 9.48 ^ab^	27.67 ± 2.07 ^bc^	21.33 ± 2.91 ^c^	55.67 ± 2.70 ^a^	46 ± 4.71 ^ab^	33 ± 3.22 ^bc^	0.0003	0.0042	0.6374
Ldh (U/L)	268.5 ± 4.74 ^c^	238.67 ± 15.46 ^cd^	215.33 ± 6.85 ^d^	394.33 ± 7.38 ^a^	316.53 ± 10.03 ^b^	269.33 ± 11.42 ^c^	<0.0001	<0.0001	0.0047

A/G: Albumin/globulin ratio, HDL: High-density lipoprotein, VLDL: Very low-density lipoprotein, LDL: Low-density lipoprotein, Ast: Aspartate aminotransferase, Alt: Alanine transaminase, Ldh: Lactate dehydrogenases. Data are expressed as the mean ± SEM, where *n* = 5. Values, or groups, with different superscript letters within a row are significantly different (*p* < 0.05) from each other based on the results of the two-way ANOVA and post-hoc multiple comparison tests.

**Table 7 biology-13-00486-t007:** Histomorphometric analysis of the muscle fiber of Nile tilapia (*Oreochromis niloticus*) reared for 8 weeks and fed on normal and high-fat fat diets supplemented with low and high doses of chitosan at both levels of dietary fat.

Parameters	F6Ch0	F6Ch5	F6Ch10	F12Ch0	F12Ch5	F12Ch10	*p* Value of Two-Way ANOVA		
							Fat	Chitosan	Interaction
Count (*n*)	567 ± 25.38 ^bc^	639.44 ± 8.32 ^ab^	653.33 ± 11.01 a	537.89 ± 28.12 ^c^	562.06 ± 23.65 ^bc^	628.89 ± 9.25 ^ab^	0.0110	0.0006	0.3399
Total area (µm^2^)	16,742.24 ± 794.06 ^b^	19,344.88 ± 250.24 ^ab^	19,899.32 ± 279.88 ^a^	15,748.05 ± 1221.6 ^bc^	16,782.73 ± 866.51 ^ab^	19,219.5 ± 228.96 ^a^	0.0235	0.0005	0.3847
Average size (µm^2^)	29.50 ± 0.32	30.25 ± 0.12	30.48 ± 0.93	29.1 ± 0.82	29.81 ± 0.50	30.58 ± 0.47	0.5344	0.0538	0.8178
Area (%)	25.79 ± 0.54 ^bc^	27.32 ± 0.27 ^ab^	27.84 ± 0.20 a	24.92 ± 0.76 c	25.56 ± 0.47 ^bc^	27.23 ± 0.26 ^ab^	0.0088	0.0004	0.4422

Values, or groups, with different superscript letters within a row are significantly different (*p* < 0.05) from each other.

## Data Availability

The authors confirm that the data supporting the findings of this study are available within the article.

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
