# Peer review of "Dietary Chitosan Attenuates High-Fat Diet-Induced Oxidative Stress, Apoptosis, and Inflammation in Nile Tilapia (Oreochromis niloticus) through Regulation of Nrf2/Kaep1 and Bcl-2/Bax Pathways"

_biology, 2024, doi:10.3390/biology13070486_

Round 1

Reviewer 1 Report

Comments and Suggestions for Authors

1.    It is recommended to include a paragraph in the introduction about previous studies of chitosan examining its effect on liver injury or other models and how it could be helpful in the study model.

2.     Please, add reference for the chitosan diet used in the study and reference for the study design (duration and % of HFD).

3.     What is the magnification of the photographs of figure 4 and figure 5? Also, include scale bar in the photographs of figure 4 and figure 5 and in the figure caption.

4.     The model used in this study is a model of lipotoxicity, so it will be better to use Oil Red O stain to confirm the lipid droplets accumulation.

5.     The discussion section includes unnecessary explanation and the histological results was repeated again in details in the discussion. Please, you should reduce this section without including unnecessary information.

6.     The tables used contain too much numbers and this is confusing and make it difficult to analyze the significance. It is recommended to highlight the parameters that were affected by HFD feeding as well as parameters that improved by chitosan diet.

7.     The visual abstract is complicated and should be simplified to indicate only the chitosan effects.

8.     What is the equivalent animal and human dose to the used chitosan dose?

Comments on the Quality of English Language

the manuscript needs revision for grammar and typing errors.

Author Response

Dear editor and respected reviewers, thank you for your time and valuable comments. Here we addressed all comments in a point-by-point response and hope we were able to explain our point of view and that our modified manuscript is suitable for publication.

Reviewer1

  1. It is recommended to include a paragraph in the introduction about previous studies of chitosan examining its effect on liver injury or other models and how it could be helpful in the study model.

Well, in this part we could not find recent research connecting chitosan to hepatic injury.

  1. Please, add reference for the chitosan diet used in the study and reference for the study design (duration and % of HFD).

The study design follows the ordinary study design for animal model experiments suing feed additives in the diet, it is our laboratory experimental design.

  1. What is the magnification of the photographs of figure 4 and figure 5? Also, include scale bar in the photographs of figure 4 and figure 5 and in the figure caption.

Corrected

  1. The model used in this study is a model of lipotoxicity, so it will be better to use Oil Red O stain to confirm the lipid droplets accumulation.

Thank you for your valuable comment, but due to shortage of fund we were unable to perform this step of staining inspite of its importance in revealing our results.

  1. The discussion section includes unnecessary explanation and the histological results was repeated again in details in the discussion. Please, you should reduce this section without including unnecessary information.

The results of the pathological investigations was explained in details in the discussion part as these results are crucial in our experiment and needed to be highlighted and compared with previous studies.

  1. The tables used contain too much numbers and this is confusing and make it difficult to analyze the significance. It is recommended to highlight the parameters that were affected by HFD feeding as well as parameters that improved by chitosan diet.

Thank you for your valuable comment. We have highlighted in bold the parameters that were deteriorated by the high-fat diet (HFD) and restored by chitosan supplementation.

  1. The visual abstract is complicated and should be simplified to indicate only the chitosan effects.

In this image we tried to explain the whole mechanism of action of Chitosan in HFD exposed fish, and we tried to make it as simple as we can but all the details must be explained to make it more useful to other researchers.

  1. What is the equivalent animal and human dose to the used chitosan dose?

The direct extrapolation of chitosan doses from fish to humans, to counteract the HFD, is not accurate due to significant species differences in metabolic rates, physiology, and dosage requirements. Fish, especially small species like Nile tilapia, have much higher metabolic rates relative to their body size compared to humans. This necessitates higher doses of compounds to achieve similar biological effects. The direct extrapolation of doses based on body surface area (BSA) or weight does not account for these metabolic differences, leading to disproportionately high HED calculations. Fish models necessitate higher doses to observe effects, which do not translate safely or effectively to human use. However, using fish models as experimental animals aims to understand the potential mechanisms and markers of chitosan's mitigation of the effects of a high-fat diet. Specifically, these models allow researchers to explore the molecular and physiological pathways influenced by chitosan, such as antioxidant defense systems, inflammatory responses, and apoptosis regulation, providing critical insights that can inform and guide subsequent human studies.

Reviewer 2 Report

Comments and Suggestions for Authors

This is an interesting study that investigated dietary chitosan on high-fat-diet-induced oxidative stress, apoptosis, and inflammation of Nile tilapia. The data are well-presented and progress logically, but several clarifications are needed, and some other comments should be considered.

1.     Could the authors clarify the purpose of the figure included on page 2? Additionally, it would be beneficial to add a legend for better understanding.

2.     What is the significance of the letters (abc) used in Tables 4-7? Additionally, it would be helpful to know if the authors have used mean +/- SD in these tables. Please clarify these points and include them in the table legends for clarity.

3.     The authors have conducted tests on non-oxidative functions, apoptosis, tight junctions, and inflammation-related gene expression using qPR-PCR. It would be beneficial if they could provide data on these gene expressions at the protein level.

Comments on the Quality of English Language

The English is satisfactory and only requires minor revisions.

Author Response

Comments and Suggestions for Authors

This is an interesting study that investigated dietary chitosan on high-fat-diet-induced oxidative stress, apoptosis, and inflammation of Nile tilapia. The data are well-presented and progress logically, but several clarifications are needed, and some other comments should be considered.

  1. Could the authors clarify the purpose of the figure included on page 2? Additionally, it would be beneficial to add a legend for better understanding.

This figure is the graphical abstract which we tried hard to explain the mechanism of action of our work in details in one figure to summarize all the important points in this study.

According to the author guidelines of the journal “Graphical abstract should represent the topic of the article in an attention-grabbing way. Moreover, it should not be exactly the same as the Figure in the paper or just a simple superposition of several subfigures”. 

  1. What is the significance of the letters (abc) used in Tables 4-7? Additionally, it would be helpful to know if the authors have used mean +/- SD in these tables. Please clarify these points and include them in the table legends for clarity.

Thanks for your valuable comment. Values, or groups, with different superscription letters within a row are significantly different (p < 0.05) from each other based on the results of the Two-way ANOVA and post-hock multiple comparison tests. We have added this note to the legends of tables. Additionally, the data for the tabulated parameters are expressed as Mean ± SEM, and the number of samples (n) is provided in the legend of each table.

  1. The authors have conducted tests on non-oxidative functions, apoptosis, tight junctions, and inflammation-related gene expression using qPR-PCR. It would be beneficial if they could provide data on these gene expressions at the protein level.

Sure this will be beneficial to provide protein data regarding all the studied genes, but due to shortage of time and fund to our work we could not proceed with proteomic analysis. But still this would be considered in our further research on the same point (HFD).